# A statistical method for removing unbalanced trials with multiple covariates in meta-analysis

**Massimo Attanasio**[1], **Fabio Aiello**[2]*, **Fabio Tinè**[3]

**1** Dipartimento di Scienze Economiche, Aziendali e Statistiche, Università di Palermo, Palermo, Italy,
**2** Facoltà di Scienze Economiche e Giuridiche, Università "Kore" di Enna, Enna, Italy, **3** Azienda Sanitaria Universitaria Giuliano Isontina (ASI GI), Trieste, Italy

* E-mail: fabio.aiello@unikore.it

**Data Availability Statement:** All relevant data are within the manuscript and its Supporting information files.

**Funding:** The research was supported by grants from Italian Ministerial grant PRIN 2017 "From

## Abstract

In meta-analysis literature, there are several checklists describing the procedures necessary to evaluate studies from a qualitative point of view, whereas preliminary quantitative and statistical investigations on the "combinability" of trials have been neglected. Covariate balance is an important prerequisite to conduct meta-analysis. We propose a method to identify unbalanced trials with respect to a set of covariates, in presence of covariate imbalance, namely when the randomized controlled trials generate a meta-sample that cannot satisfy the requisite of randomization/combinability in meta-analysis. The method is able to identify the unbalanced trials, through four stages aimed at achieving combinability. The studies responsible for the imbalance are identified, and then they can be eliminated. The proposed procedure is simple and relies on the combined Anderson-Darling test applied to the Empirical Cumulative Distribution Functions of both experimental and control meta-arms. To illustrate the method in practice, two datasets from well-known meta-analyses in the literature are used.

## Introduction

Meta-analysis is an analytical technique designed to "combine" findings from multiple studies. It is commonly used to evaluate studies about medical interventions with the aim to provide researchers, policymakers, and clinicians with useful information. Combinability is a technique that integrates data obtained from dissimilar studies. In meta-analysis' literature, combinability is defined as "*the extent to which separate studies are similar enough*" [1], or "*the extent to which separate studies measure the same thing*" [2]. Of significant interest is the scientific process that enables the integration of studies with similar outcomes. Several approaches have been developed to offer rationale and provide procedures on how studies are chosen, how the data are assembled and how the results are reported.

The literature is vast, and several guidelines have been proposed. The most popular guidelines are QUORUM [3, 4], *Quality of Reporting of Meta-Analyses* and PRISMA [5], *Preferred Reporting Items of Systematic reviews and Meta-Analyses*, a technique that has evolved from Quorum. These guidelines provide a checklist that facilitates a "good" meta-analysis or systematic review.

high school to job placement: micro-data life course analysis of university student mobility and its impact on the Italian North-South divide.", n. 2017HBTK5P, of which MA is Principal Investigator. The funders had no role in study design, data collection and analysis, decision to publish, or preparation of the manuscript.

**Competing interests:** The authors have declared that no competing interests exist.

The PRISMA checklist consists in qualitative issues and does not cover quantitative issues. Its statistical recommendations focus exclusively with *effect size* measures, confidence intervals and with measures that assess heterogeneity, subgroup analysis or other sources of biases, for instance publication bias. Yusuf and Pogue [6] have already stressed how small sample trial meta-analyses are more susceptible to bias and have advised to choose large sample meta-analyses, to obtain more reliable answers and explore interactions among subgroups. additionally, Cochrane Collaboration [7] defines "systematic review as a review of a clearly formulated question that uses systematic and explicit methods to identify, select, and critically appraise relevant research hypotheses to collect and analyze data from the studies that are included in the review".

To summarize, while qualitative issues of combinability are always examined extensively [8], quantitative issues are essentially limited to sample sizes and effect sizes. In reality, the quantitative assessment of clinical combinability studies is unsatisfactory [9, 10], because it lacks specific quantitative criteria to establish when trials can be considered similar enough. That is why here we propose a method to detect the trials responsible for the lack of combinability, i.e., imbalance between the treatment groups, with respect to some potential risk factors.

In a single randomized controlled trial (RCT), covariate imbalance is a very important statistical problem that has been investigated in many scientific papers. Overall, in a single RCT covariate balance occurs when the patients in each group of treatment are similar as close as possible, particularly with regard to prognostic factors [11–13]. When this condition does not hold, then it is referred as covariate imbalance. The issue arises when dissimilarity between the experimental (*exp*) and control (*ctrl*) arms due to covariate imbalances violates the assumption of "combinability", which is a fundamental premise of meta-analysis. Meta-analysis operates on the assumption that, during the random allocation process to the *exp* and *ctrl* arms, the expected level of covariate distribution imbalance should ideally be zero [14]. Covariate balance is not always assessed before conducting a meta-analysis, automatically assuming that individual studies are well-balanced. However, it can happen that some studies do not exhibit covariate imbalance for some or all covariates, or, as Trowman et al., [15] have pointed out that a meta-analysis imbalance may not result just from a baseline imbalance of one trial, but rather from a cumulative effect of smaller imbalances. In both scenarios, the meta-analysis present covariate imbalance. Other scholars have dealt with covariate imbalance. Riley et al. [16] and Ciolino et al. [14] present methods to assess continuous baseline covariate imbalance across treatment groups in clinical trials with a continuous outcome; Clark et al. [17, 18] claim the relevance of bias due to covariate imbalance in meta-analysis studies with respect to allocation concealment; Hicks et al. [19] and Wewege et al. [20] consider baseline imbalance through statistics calculated for each covariate and they remove those studies where differences are not acceptable.

Here, we support the proposal by Trowman et al. [15] that single slightly unbalanced RCTs could generate a meta-sample that cannot satisfy the randomization/combinability requisite of meta-analysis. The upshot is a "rule of thumb" procedure aimed at eliminating unbalanced trials, which cannot be applied when the number of trials involved is large. Alternatively, individual patient data (IPD) may be used instead of meta-analysis, with the caveat that it requires collection of data of all patients involved in all relevant studies [21]. In addition, if a significant portion of the trials included in a systematic review have baseline imbalance, then combining them in a meta-analysis will produce a misleading result [15]. Thus, to mitigate such bias, it is crucial to conduct meta-regressions with balanced trials.

In this context, we propose a method to identify the studies responsible for the imbalance, with respect to a set of covariates. The Proposed statistical method section describes the main

stages of the method to assess the covariate balance in meta-analysis. The Two datasets section illustrates the two meta-analysis datasets used in our application, coming from the Cholesterol Treatment Trialist' (CTT) Collaboration and the Cochrane library. The Notation section defines the objects and the abbreviations used in the paper. The section Application to the two datasets illustrates how the method is applied, presenting the results and employing a logit model for investigating the relation between balanced and unbalanced trials. In the section Conclusion, we discuss concluding points.

## The proposed statistical method

This paper starts from the results of a previous work [22], where a tool was developed to assess the covariate imbalance with respect to a single covariate. Recognizing that clinical practice always involves multiple covariates, we now propose a method for detecting unbalanced trials. The method proposed in this work has:

1. extended the combinability procedure in the presence of three covariates, considering that clinical studies often involve more than one covariate. This also led to a generalization of the test statistics used (for a better understanding of this aspect, changes have been made in the introduction),

2. introduced a new section in this work, a kind of ex-post verification, dedicated to estimating the effect size with unbalanced and balanced trials,

3. included a simulation in the S1 Appendix, providing additional strength to the procedure.

We propose a stepwise procedure for assessing the imbalance between the treatment groups, with respect to potential factors, comparing their distributions in the treatment groups, without any assumption on their shapes. We classify the potential factors of imbalance as:

- study-level variables (SLVs), which usually include design variables, or population structure variables,

- patient-level variables (PLVs), which are all the baseline variables related to the patients.

To illustrate this new method, we will refer to the objects defined in S2 Appendix, which are:

- the *exp* meta-arm, i.e., representing a collection of similar experimental arms,

- the *ctrl* meta-arm, i.e., representing a collection of similar control arms,

- the Empirical Cumulative Distribution Function (ECDF) of a PLV built for each meta-arm (see Table 1 in S2 Appendix), consisting in a distribution function in which the frequencies are replaced by the sum of the sample sizes of the arms for each PLV value.

The covariate balance holds if the ECDFs are not statistically different.

The rationale of the method is to assess the combinability, to identify the studies responsible for the imbalance. Overall, the method here proposed adheres to four sequential stages.

### Assessing the marginal combinability

Marginal combinability holds when the randomization process holds with respect to some basic prognostic factors, over all the levels of a given SLV, that is, the PLVs' ECDFs are not statistically different, controlling for the SLV levels [22]. This is investigated both graphically and analytically, through the Anderson-Darling test (see the Notation section).

### Assessing the basic combinability

Basic combinability holds when the PLVs' ECDFs for each meta-arm are not statistically different. If basic combinability does not hold, the meta-analysis cannot be conducted without intervention and/or correction [22]. Also in this case, the basic combinability is investigated both graphically and analytically, through the Anderson-Darling test.

### Identifying the unbalanced trials

Among the original studies included in a meta-analysis, an iterative procedure is employed to identify the studies responsible for the highest observed imbalance. This process continues until a statistically reasonable balance is achieved between the *exp* and *ctrl* meta-arms. To do this, we establish an iterative procedure based on a pooled quantity over the PLVs, capable of detecting the trials responsible for the imbalances. Once identified the unbalanced trials, it is necessary to remove these trials.

### Removing the unbalanced trials

In this case, it is important to consider both qualitative and quantitative criteria (which are not strictly statistical evaluations). Regarding qualitative issues, the eliminated studies should not compromise the meta-analysis because the studies that have an important scientific value cannot be omitted unless one re-defines the meta-analysis parameters. This can happen if one eliminates the studies that represent a specific subgroup (for example the geographic areas, specific dosages, or important subcases such as diabetics, etc.). Otherwise, the objectives of the meta-analysis should be redefined. Instead, quantitatively, one must balance the total number of eliminated studies and the number of patients corresponding to those studies. In fact, from a practical standpoint it would be best not to surpass a convenient limit for the number of studies, or the number of patients eliminated.

### The two datasets

The two examples used pertain to studies with higher incidence rates. The first dataset (S3 File), hereafter named *Chol* (Table 1), is drawn from a well-known meta-analysis [23]. It comprises 26 multicentric randomized trials, conducted by the Cholesterol Treatment Trialist' (CTT) Collaboration, involving two types of trials: more intensive statin regimens versus less intensive statin regimens (5 trials) and statin versus control comparisons (21 trials). We selected the 21 trials of the second type (the PRISMA flowchart is depicted in Fig 1).

These trials involve 129,144 participants with treatment durations of at least 2 years for statin (*exp*) versus control (*ctrl*), assessing the efficacy and safety of cholesterol-lowering therapy on the risk of occlusive vascular events in a wide range of individuals [24–44]. The median follow-up is 4.8 years, during which 7136 participants out of 64540 participants (2.8% per annum) allocated to statin therapy experienced their first major vascular events, compared to 8934 out of 64604 participants (3.6% per annum) participants in the control group. In the *Chol* dataset, the trials were first classified as European, mostly European, North American, Australian, Japanese, and multi-continental. Subsequently, we combined the first two into the European category and grouped the others as non-European.

The second dataset (S4 File), hereafter named *Hep* (S1 Table), is derived from a Cochrane review on Hepatitis C [45]. The review commenced with 72 studies, of which 32 were excluded, based on various criteria (the PRISMA flowchart is provided in Fig 2). We identified 40 studies [46–85], involving 2999 patients in the *ctrl* arms and 4108 in the *exp* arms, conducted in Europe and North America.

**Table 1. Cholesterol Treatment Trialists' (CTT) Collaborators 21 studies (*Chol* dataset): Selected SLV and PLVs.**

| | | Arm | | | | | | | | |
|---|---|---|---|---|---|---|---|---|---|---|
| | | Ctrl | | | | | Exp | | | | |
| Trial name[a] | SLV | No. of patients | No. of any major vascular event | PLVs | | | No. of patients | No. of any major vascular event | PLVs | | |
| | Continent | | | *mean (age)* | *p(diab)* | *p(male)* | | | *mean (age)* | *p(diab)* | *p(male)* |
| SSSS | European | 2223 | 796 | 58.60 | 0.04 | 0.81 | 2221 | 555 | 58.60 | 0.05 | 0.82 |
| WOSCOPS | European | 3293 | 318 | 55.10 | 0.01 | 1.00 | 3302 | 232 | 56.30 | 0.01 | 1.00 |
| CARE | North AM | 2078 | 553 | 59.00 | 0.15 | 0.86 | 2081 | 433 | 59.00 | 0.14 | 0.86 |
| Post-CABG | North AM | 677 | 100 | 61.60 | 0.09 | 0.91 | 674 | 79 | 61.40 | 0.09 | 0.93 |
| AFCAPS | North AM | 3301 | 201 | 58.00 | 0.05 | 0.85 | 3304 | 143 | 58.00 | 0.07 | 0.85 |
| LIPID | Australia | 4502 | 1153 | 62.00 | 0.09 | 0.83 | 4512 | 936 | 62.00 | 0.09 | 0.83 |
| GISSI-P | European | 2133 | 231 | 60.00 | 0.14 | 0.86 | 2138 | 208 | 59.70 | 0.13 | 0.86 |
| LIPS | Mostly EU | 833 | 195 | 60.00 | 0.10 | 0.83 | 844 | 164 | 60.00 | 0.14 | 0.84 |
| HPS | European | 10267 | 2043 | 65.20 | 0.29 | 0.75 | 10269 | 1511 | 65.20 | 0.29 | 0.75 |
| PROSPER | European | 2913 | 495 | 75.30 | 0.11 | 0.48 | 2891 | 431 | 75.40 | 0.11 | 0.48 |
| ALLHAT-LLT | North AM | 5185 | 812 | 66.30 | 0.34 | 0.51 | 5170 | 758 | 66.40 | 0.36 | 0.51 |
| ASCOT-LLA | European | 5137 | 307 | 63.20 | 0.25 | 0.81 | 5168 | 217 | 63.10 | 0.24 | 0.81 |
| ALERT | Mostly EU | 1052 | 140 | 50.00 | 0.19 | 0.65 | 1050 | 135 | 49.50 | 0.19 | 0.67 |
| CARDS | European | 1410 | 123 | 61.80 | 1.00 | 0.68 | 1428 | 81 | 61.50 | 1.00 | 0.68 |
| ALLIANCE | North AM | 1225 | 293 | 61.30 | 0.21 | 0.82 | 1217 | 254 | 61.10 | 0.23 | 0.82 |
| 4D | European | 636 | 162 | 65.70 | 1.00 | 0.54 | 619 | 144 | 65.70 | 1.00 | 0.54 |
| ASPEN | Multi-Continent | 1199 | 136 | 61.00 | 1.00 | 0.67 | 1211 | 114 | 61.10 | 1.00 | 0.66 |
| MEGA | Japan | 3966 | 140 | 58.40 | 0.21 | 0.31 | 3866 | 102 | 58.20 | 0.21 | 0.32 |
| JUPITER | Multi-Continent | 8901 | 194 | 66.00 | 0.00 | 0.62 | 8901 | 105 | 66.00 | 0.00 | 0.61 |
| GISSI-HF | European | 2289 | 174 | 68.00 | 0.25 | 0.79 | 2285 | 172 | 68.00 | 0.27 | 0.76 |
| AURORA | Multi-Continent | 1384 | 368 | 64.30 | 0.25 | 0.65 | 1389 | 362 | 64.10 | 0.28 | 0.61 |
| | Totals | 64604 | 8934 | | | | 64540 | 7136 | | | |

[a]Trial names are consistent with the work of CCT.

As already said, we aim to evaluate the combinability of the studies in a metanalysis, with respect to the PLVs, considering two different types of balance [22]. The first type regards the combinability of the trials concerning the levels of the SLVs (marginal combinability). The second type regards the combinability of the trials with respect to the treatment, *exp* or *ctrl* (basic combinability).

## Notation

To avoid cumbersome notation, we have introduced the following terminology:

1. Let $\mathbf{S} = \{S_1, S_2, \ldots, S_I\}$ be the original set of $I$ trials, with cardinality $|\mathbf{S}| = I$, collected for the meta-analysis. Each of the $I$ trials, $S_i$ (for $i = 1, 2, \ldots, I$), has at least two ($k$) arms, the control ($k = 1$) arm and the experimental ($k = 2$) arm (see S1 Appendix).

2. Let $\mathbf{S}_{(-i)} = \{\mathbf{S}\backslash\{S_i\}\}$, for $i = 1, 2, \ldots, I$, be a set of $I-1$ trials. In this way, we get $I$ sets of this kind, each with cardinality $|I-1|$.

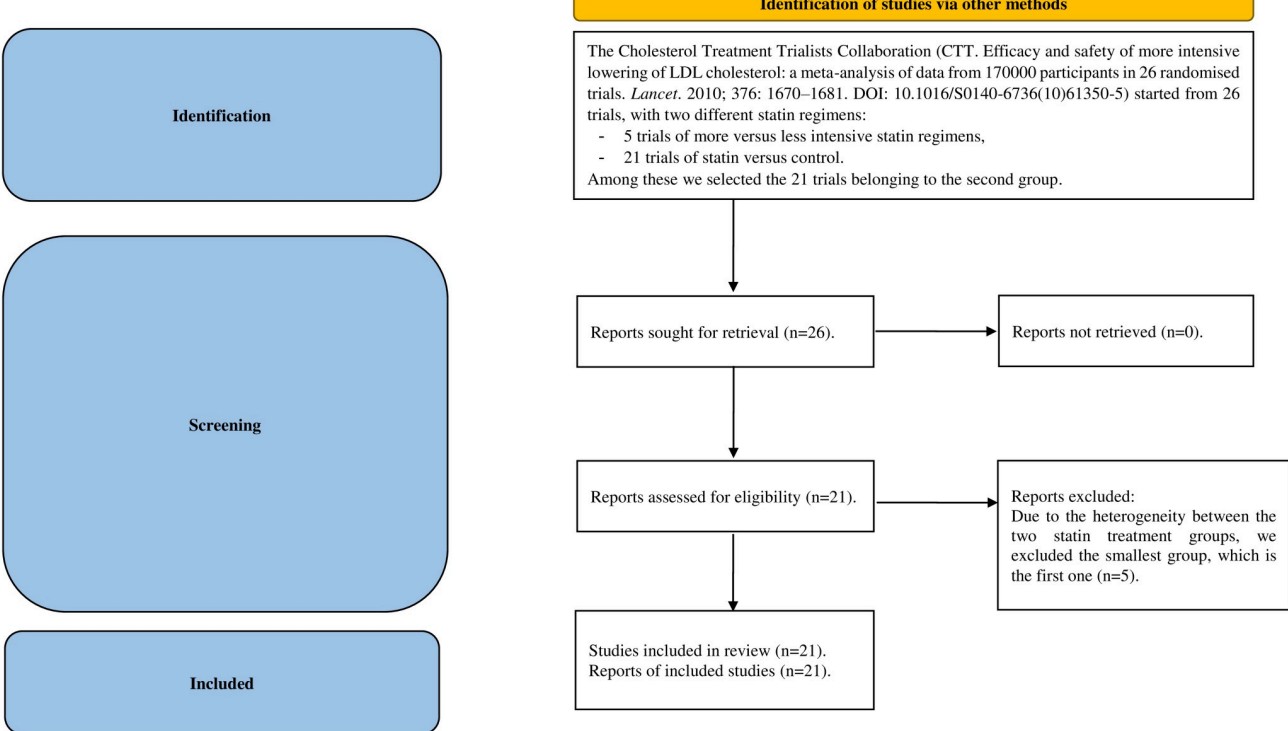

**Fig 1. The PRISMA flowchart of *Chol* dataset.** *Consider, if feasible to do so, reporting the number of records identified from each database or register searched (rather than the total number across all databases/registers). **If automation tools were used, indicate how many records were excluded by a human and how many were excluded by automation tools. *From*: Page MJ, McKenzie JE, Bossuyt PM, Boutron I, Hoffmann TC, Mulrow CD, et al. The PRISMA 2020 statement: an updated guideline for reporting systematic reviews. BMJ 2021; 372: n71. doi: 10.1136/bmj.n71. For more information, visit: http://www.prisma-statement.org/.

3. Let $\mathbf{S}_{(-i)(-i')} = \{\mathbf{S}\backslash\{S_i, S_{i'}\}\}$, for $i' = 1, 2, \ldots, I-1$, be a set of $I-2$ trials. In this way, we get $I-1$ sets of this kind, each with cardinality $|I-2|$.

4. So forth for the triples, until $\mathbf{S}(all)_{(-i)(-i')\ldots(-(I-1))}$.

5. Let $\mathbf{ctrl}$, $\mathbf{ctrl}_{(-i)}$, $\mathbf{ctrl}_{(-i)(-i')}$ be the control meta-arms built over the sets $\mathbf{S}$, $\mathbf{S}_{(-i)}$, $\mathbf{S}_{(-i)(-i')}$.

6. Let $\mathbf{exp}$, $\mathbf{exp}_{(-i)}$, $\mathbf{exp}_{(-i)(-i')}$ be the experimental meta-arms built over the sets $\mathbf{S}$, $\mathbf{S}_{(-i)}$, $\mathbf{S}_{(-i)(-i')}$.

7. $\mathbf{PLV} = \{PLV_1, PLV_2, \ldots, PLV_H\}$ (for $h = 1, 2, \ldots, H$) be a given vector of $H$ PLVs.

8. Let $\mathrm{T}\left(A_{hkN}^2\right)$ be the $k$-sample Anderson-Darling test defined as:

$$\mathrm{T}\left(A_{hkN}^2\right) = \frac{A_{hkN}^2 - \mu_{hN}}{\sigma_{hN}} \qquad (1)$$

where $N = m + n$, that are the sample sizes of the two meta-arms, the $A_{hkN}^2$ is the $k$-sample Anderson-Darling criterion, computed for the $k$ meta-arms and for an individual $h$th PLV, and where $\mu_{hN} = k-1$ and $\sigma_{hN}$ are the mean and the standard deviation of $A_{hkN}^2$. Details on the statistical distributions are included in [86].

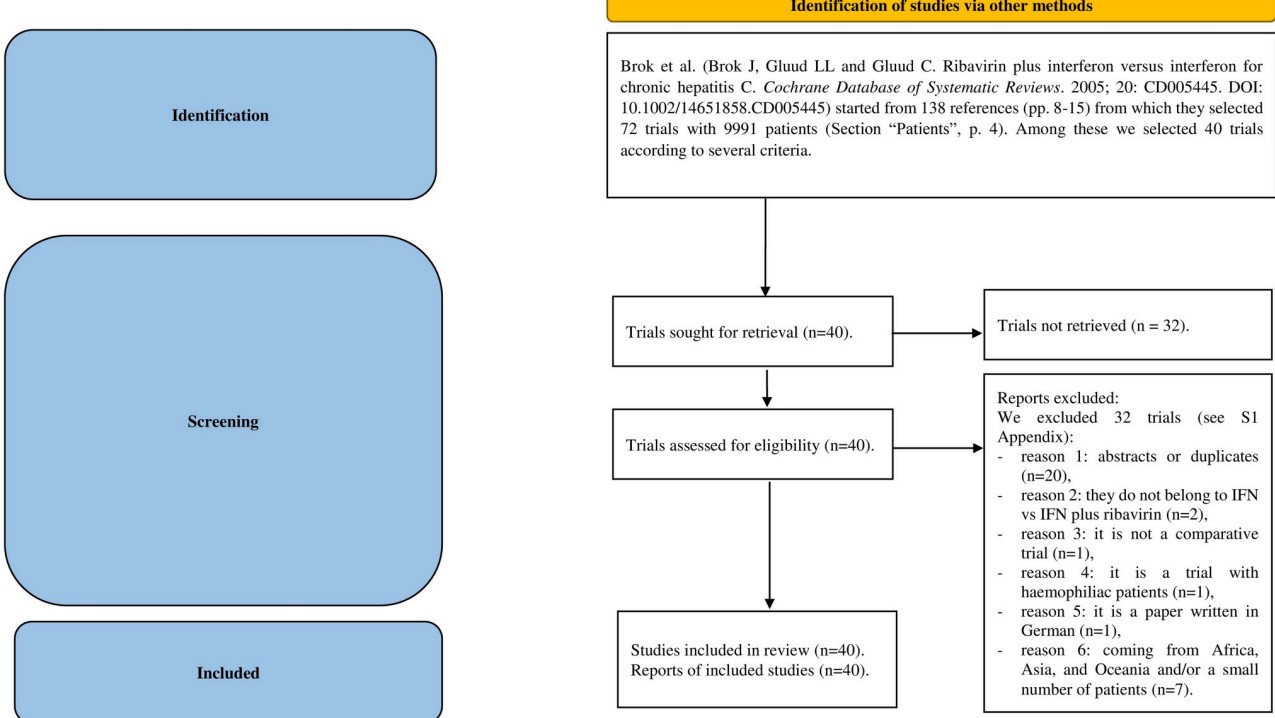

**Fig 2. The PRISMA flowchart of *Hep* dataset.** *Consider, if feasible to do so, reporting the number of records identified from each database or register searched (rather than the total number across all databases/registers). **If automation tools were used, indicate how many records were excluded by a human and how many were excluded by automation tools. *From*: Page MJ, McKenzie JE, Bossuyt PM, Boutron I, Hoffmann TC, Mulrow CD, et al. The PRISMA 2020 statement: an updated guideline for reporting systematic reviews. BMJ 2021; 372: n71. doi: 10.1136/bmj.n71. For more information, visit: http://www.prisma-statement.org/.

9. Let $A_c^2 = \sum_{h=1}^{H} A_{hkN}^2$ be the combined Anderson-Darling criterion, always computed for the $k$ meta-arms, summing the $A_{hkN}^2$ criteria over all PLVs. Hence, the overall test is given by:

$$T\left(A_c^2\right) = \frac{A_c^2 - \mu_c}{\sigma_c} \qquad (2)$$

where $\mu_c = \sum_{h=1}^{H} \mu_{hN}$ and $\sigma_c = \sqrt{\sum_{h=1}^{H} \sigma_{hN}^2}$ are the mean and the standard deviation of $A_c^2$. The $T\left(A_c^2\right)$ statistic is the combined Anderson-Darling $k$-sample test under the hypothesis that the independent arms within each trial come from a common unspecified continuous distribution.

In meta-analysis, all arms of all trials are assumed to be independent and from identical continuous distributions. Both the individual criterion, $A_{hkN}^2$, and the combined Anderson-Darling criterion, $A_c^2$, (and the corresponding standardized statistics, $T\left(A_{hkN}^2\right)$ and $T_c$, respectively) are used to test simultaneously whether the arms of each trial come from the same continuous distribution function, i.e., whether they are balanced. These standardized statistics are expected to be zero under the null hypothesis. Thus, the larger the statistics, the greater the overall imbalance.

In the case of dependent samples, one can refer to the suggestions made by Lin and Sullivan [87] and Han et al. [88].

## The application to the two datasets

This section demonstrates the application of the proposed method. The first two stages are applied to both datasets, but for brevity, we will illustrate the iterative procedure only for the *Chol* dataset, while the results will be reported for both datasets.

The *Chol* and *Hep* datasets comprise of 21 and 40 trials, respectively. In both datasets, we refer to the SLV *Continent* (European, EU, and Non-European, Non-EU), because it reflects different epidemiological populations (S2 Table). The PLVs consist of well-known risk factors associated to the disease under study. In the *Chol* dataset, these include the patients' mean age, *mean*(*age*), the proportion of diabetics, *p*(*diab*), the proportion of males, *p*(*male*). In the *Hep* dataset, the PLVs include the patients' mean age, *mean*(*age*), the proportion of cirrhotic patients, *p*(*cirr*), and the proportion of males, *p*(*male*). We applied the method in three stages as described above.

### Assessing the marginal combinability

We construct the ECDFs of each PLV, controlling for the two levels of the SLV *Continent*, i.e., European, and Non-European trials. We then compared each pair of ECDFs using the Anderson-Darling test for both datasets. Fig 3 illustrates that the ECDFs are noticeably different from each other, with all the *p*-values being significant. Therefore, the distributions of the PLVs are structurally and statistically different in European and Non-European trials. For brevity, we applied our method only to the subsets of European studies for both datasets: **S1** (with |**S1**| = 11) for the *Chol* dataset, and **S2** (with |**S2**| = 34) for the *Hep* dataset.

### Assessing the basic combinability

We constructed the ECDFs of each PLV separately for both the *exp* and *ctrl* meta-arms within the **S1** and **S2** subsets of the *Chol* and *Hep* datasets, respectively. We then investigated the basic combinability of data by comparing all pairs of ECDFs, both graphically (Fig 4) and analytically (Table 2), using the Anderson-Darling statistics based on the quantities $A^2_{hkN}$, $\mu_{hN}$, $\sigma_{hN}$.

The ECDFs for the first dataset (see Fig 4a–4c) exhibit closeness, while in the second dataset (see Fig 4d–4f) the ECDFs are less similar. This difference is likely due the varied distribution among the studies in the datasets.

Table 2 reports the quantities, namely, $A^2_{hkN}$, $\mu_{hN}$, $\sigma_{hN}$, to compute the individual and the combined Anderson-Darling statistics, $T(A^2_{hkN})$ and $T_h$ (for $h = 1, 2, 3$), which measure the basic imbalance concerning the PLVs. The overall $T_c$ is obtained by summing the three $T(A^2_{hkN})$. The *p*-values are all significant, denoting that the ECDFs are all statistically different and hence the trials are not balanced concerning the PLVs under consideration.

### Identifying the unbalanced trials

We developed a backward reduction procedure to select the balanced trials, implemented using the "kSample" package of R software [89]. It compares the ECDFs of the meta-arms using the A-D test. To avoid ties in the ECDFs, the values were perturbed by a random component. The procedure identifying the unbalanced studies is based on comparing the quantities $T_{c(-i)}$, calculated over the subsets:

$$\mathbf{S1}_{(-i)} = \{\mathbf{S1}\backslash\{S1_i\}\} \Rightarrow T_{c(-i)} \text{ where } |\mathbf{S1}_{(-i)}| = 10, \forall\, i = 1, \ldots, 11.$$

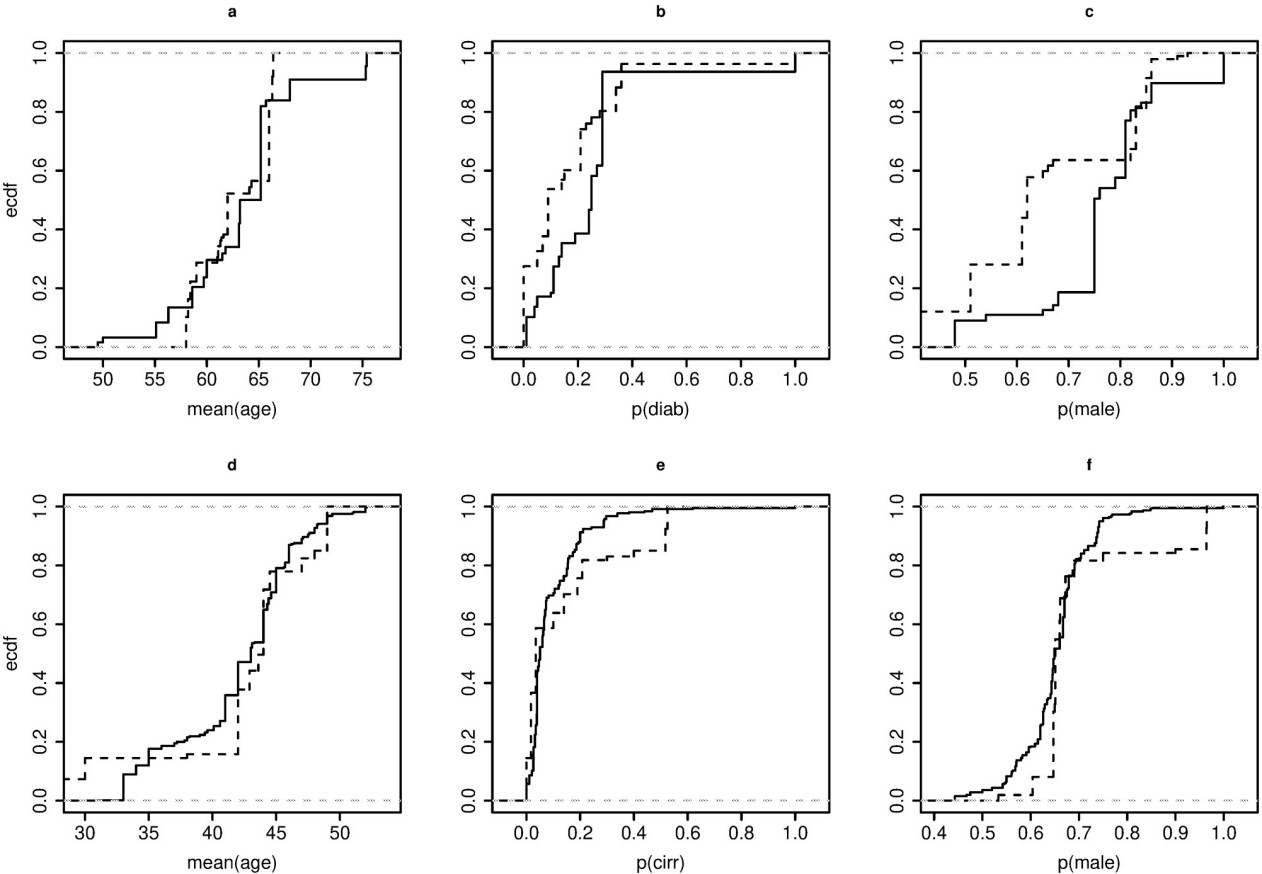

**Fig 3. ECDFs of the European (——) and Non-European (– –) meta-arms, with respect to the PLVs.** *Chol* dataset (a, b, c); *Hep* dataset (d, e, f).

$$\mathbf{S2}_{(-j)} = \{\mathbf{S2} \setminus \{\mathrm{S2}_j\}\} \Rightarrow \mathrm{T}_{c(-j)} \text{ where } |\mathbf{S2}_{(-j)}| = 33, \forall\, j = 1, \ldots, 34.$$

For simplicity, let us assume that all the risk factors have equal weight (although different weights can also be applied) and we will proceed with the subset **S1** of the *Chol* dataset, even though the results will be reported for both datasets.

At each step, the decision rule eliminates the study associated with the greatest overall imbalance from the initial set, **S1**. The steps of the iterative procedure are:

1. Build $I$ sets $\{\mathbf{S1} \setminus \{\mathrm{S1}_i\}\}$, for $i = 1, 2, \ldots, I$, whose cardinality is $|I{-}1|$.

2. Compute the quantities $\mathrm{T}_{h(-i)}$, for $h = 1, 2, 3$, and $\{\mathrm{T}_{c(-i)}\}$, $\forall\, i = 1, \ldots, I$.

3. Identify the $min_i\{\mathrm{T}_{c(-i)}\}$, and then the corresponding $i$th study.

4. Remove the $i$th study, $\mathrm{S1}_i$, and consider the set $\mathbf{S1}_{(-i)} = \{\mathbf{S1} \setminus \{\mathrm{S1}_i\}\}$.

5. Rename $\mathbf{S1} = \{\mathbf{S1} \setminus \{\mathrm{S1}_i\}\}$.

6. If $min_i\{\mathrm{T}_{c(-i)}\}$ is not significant at the level of $\alpha = 0.05$, then stop,

7. Otherwise, go to step 1.

The procedure terminates when the value $min_i\{\mathrm{T}_{c(-i)}\}$ is not significant, and the "latest" **S1**, consisting of the non-removed trials, is *balanced* with respect to the chosen covariates.

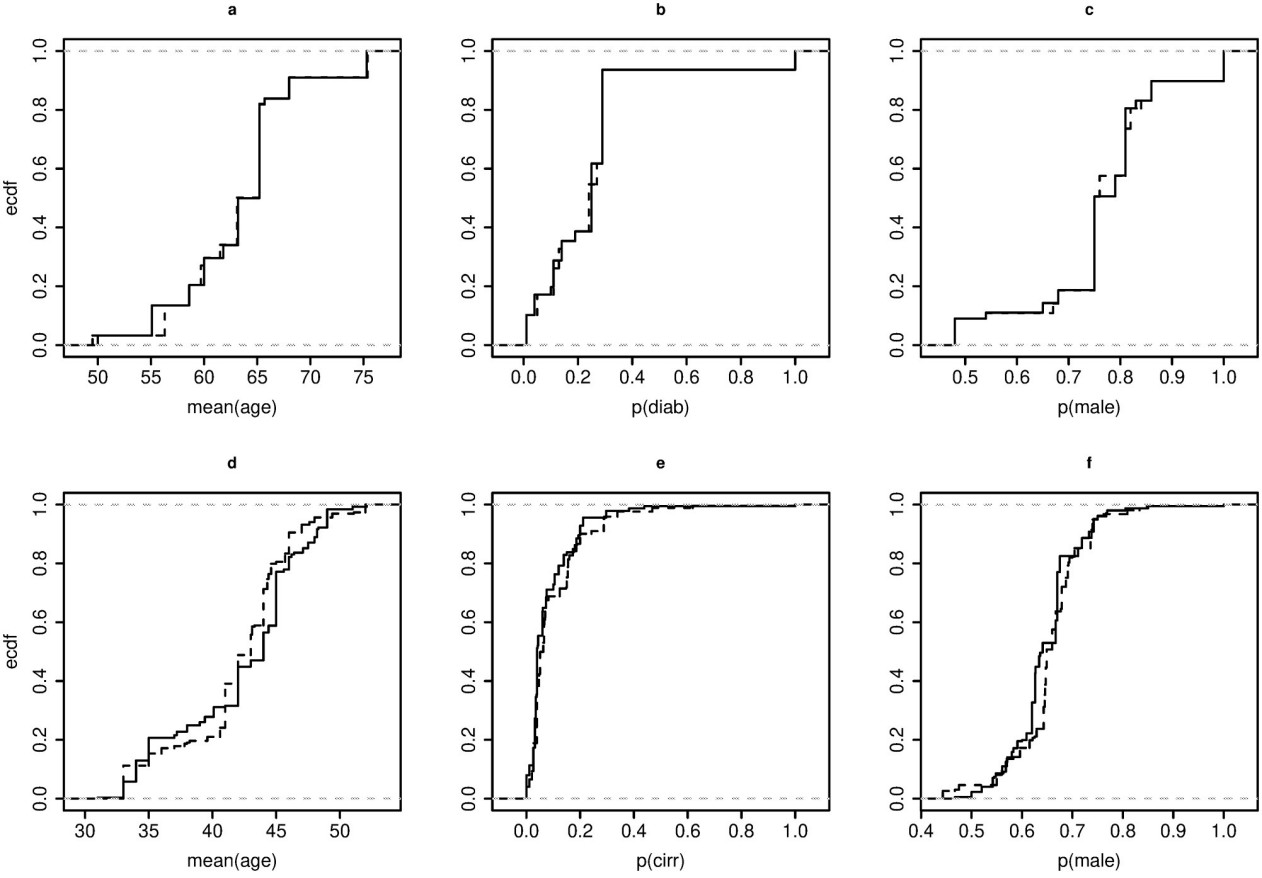

**Fig 4. ECDFs of the *exp* (——) and *ctrl* (——) meta-arms in the European studies, for the PLVs. S1** (a, b, c) of the *Chol* dataset and **S2** (d, e, f) of the *Hep* dataset.

### 1ˢᵗ iteration of the procedure.

1. Build 11 sets $\mathbf{S1}_{(-i)} = \{\mathbf{S1}\backslash\{S1_i\}\} \;\forall\; i = 1, \ldots, 11$, whose cardinality is 10.

2. Compute $T_{1(-i)}$, $T_{2(-i)}$, $T_{3(-i)}$, and $T_{c(-i)}$ (S3 Table).

3. As the $min_i\{T_{c(-i)}\}$ is $T_{1(-12)} = 58.9$, then identify $S1_{12}$.

4. Remove $S1_{12}$; consider $\mathbf{S1}_{(-12)} = \{\mathbf{S1}\backslash\{S1_{12}\}\}$.

**Table 2. Quantities of the Anderson-Darling statistics and *p*-values, for S1 (*Chol* dataset) and S2 (*Hep* datasets).**

|  | S1 | | | | S2 | | | |
|---|---|---|---|---|---|---|---|---|
|  | *mean(age)* | *p(diab)* | *p(male)* | *Combined* | *mean(age)* | *p(cirr)* | *p(male)* | *Combined* |
| $A^2_{hkN}$ | 11.09 | 51.42 | 25.29 | 87.81 | 158.20 | 2450.00 | 1086.00 | 3694.20 |
| $\mu_{hN}$ | 1 | 1 | 1 | 3 | 1 | 1 | 1 | 3 |
| $\sigma_{hN}$ | 0.761 | 0.761 | 0.761 | 1.319 | 0.761 | 0.761 | 0.761 | 1.319 |
| $T(A^2_{hkN})$ | 13.25 | 66.22 | 31.91 | – | 206.50 | 3218.00 | 1425.00 | – |
| $T_c$ | – | – | – | 64.30 | – | – | – | 2798.48 |
| *p* | 0.000 | 0.000 | 0.000 | 0.000 | 0.000 | 0.000 | 0.000 | 0.000 |

5. Rename **S1** = {**S1**\{$S1_{12}$}}.

6. Since $T_{c(-12)}$ is significant, return to step 1.

### 2nd iteration of the procedure.

1. Build 10 sets $\mathbf{S1}_{(-i)} = \{\mathbf{S1}\backslash\{S1_i\}\} \ \forall \ i = 1, \ldots, 10$, whose cardinality is 9.

2. Compute $T_{1(-i)}$, $T_{2(-i)}$, $T_{3(-i)}$, and $T_{c(-i)}$ (S4 Table).

3. As the $min_i\{T_{c(-i)}\}$ is $T_{c(-1)} = 43.1$, then identify $S1_1$.

4. Remove $S1_1$; consider $\mathbf{S1}_{(-1)} = \{\mathbf{S1}\backslash\{S1_1\}\}$.

5. Rename **S1** = {**S1**\{$S1_1$}}.

6. Since $T_{c(-1)}$ is significant, return to step 1.

Now, let's skip to the last one, keeping in mind that we removed 5 studies ($S1_{12}$, $S1_1$, $S1_2$, $S1_8$, $S1_{20}$).

### 6th iteration.

1. Build 6 sets $\mathbf{S1}_{(-i)} = \{\mathbf{S1}\backslash\{S1_i\}\} \ \forall \ i = 1, \ldots, 6$, whose cardinality is 5.

2. Compute $T_{1(-i)}$, $T_{2(-i)}$, $T_{3(-i)}$, and $T_{c(-i)}$ (S5 Table).

3. As the $min_i\{T_{c(-i)}\}$ is $T_{c(-7)} = 1.1$, then identify $S1_7$.

4. Remove $S1_7$; consider $S1_{(-7)} = \{\mathbf{S1}\backslash\{S1_7\}\}$.

5. Rename **S1** = {**S1**\{$S1_7$}}.

6. Since $T_c(-7)$ is not significant ($p = 0.138$), stop the procedure.

Table 4 summarizes the results of the backward reduction procedure at each step, for both datasets, *Chol* and *Hep*. The iterations are 6 for **S1**, and 4 for **S2**, leading to the final subset of balanced trials:

- **BAL1** = ($S1_9$, $S1_{10}$, $S1_{13}$, $S1_{14}$, $S1_{16}$), for *Chol* dataset.

- **BAL2** = ($S2_1$, $S2_2$, $S2_4$, $S2_5$, $S2_6$, $S2_7$, $S2_9$, $S2_{10}$, $S2_{11}$, $S2_{12}$, $S2_{13}$), for *Hep* dataset.

   the unbalanced trials are:

- **UNB1** = **S1** −**BAL1** for *Chol* dataset.

- **UNB2** = **S2** −**BAL2** for *Hep* dataset.

   The last columns of Table 3 report, for both datasets, the percentage of lost patients at each iteration, which reaches 49.5% and 24.3%, respectively.

   As expected, the ECDFs built over **BAL1** and **BAL2** show reasonable overlapping between the *exp* and *ctrl* meta-arms (Fig 5).

## The effect of the *imbalance* on the outcome variable

The previous procedure effectively identifies trials that may be responsible for imbalances between *exp* and *ctrl* meta-arms, without taking into account the outcome variables. The proposed solution aims to remove the unbalanced trials to conduct a "proper" meta-analysis. However, in this section we apply a meta-regression to evaluate the effects of the treatment (i.e., the arm type) and the "imbalance" on the outcomes of the *Chol* and *Hep*

**Table 3. Results of the backward reduction procedure by iteration: Statistics, *p*-values, and reduction of both studies and patients.** S1 (*Chol* dataset) and S2 (*Hep* dataset).

| Iteration $r$ | S1 | | | | | | | S2 | | | | | | |
|---|---|---|---|---|---|---|---|---|---|---|---|---|---|---|
| | $T_c$ statistics | $p$ | Deleted Study | Deleted pts | No. of studies | No. of pts | Pts' Reduction (%) | $T_c$ statistics | $p$ | Deleted Study | Deleted pts | No. of studies | No. of pts | Pts' Reduction (%) |
| 0 | 64.3 | < 0.001 | 0 | 0 | 11 | 64401 | 0 | 2798.5 | < 0.001 | 0 | 0 | 34 | 6473 | 0 |
| 1 | 59.8 | 0.008 | $S1_{12}$ | 10305 | 10 | 54096 | -16.0 | 14.9 | < 0.001 | $S2_3$ | 303 | 33 | 6170 | -4.7 |
| 2 | 43.1 | < 0.001 | $S1_1$ | 4444 | 9 | 49652 | -22.9 | 6.2 | < 0.001 | $S2_{25}$ | 832 | 32 | 5338 | -17.5 |
| 3 | 18.7 | < 0.001 | $S1_{20}$ | 4574 | 8 | 45078 | -30.0 | 3.3 | < 0.05 | $S2_{18}$ | 376 | 31 | 4962 | -23.3 |
| 4 | 13.8 | < 0.001 | $S1_8$ | 1677 | 7 | 43401 | -32.6 | 1.8 | 0.061 | $S2_8$ | 60 | 30 | 4902 | -24.3 |
| 5 | 10.5 | < 0.001 | $S1_2$ | 6595 | 6 | 36806 | -42.8 | - | - | - | - | - | - | - |
| 6 | 1.1 | 0.138 | $S1_7$ | 4271 | 5 | 32535 | -49.5 | - | - | - | - | - | - | - |

dataset. The goal of this application is to illustrate the effect that including unbalanced trials would have had on the outcome. The outcomes are the probability *p* of "occlusive vascular events" for the *Chol* dataset, and the "sustained response" for the *Hep* dataset. To achieve this, we will use a meta-regression logit model with two dummy variables, defined

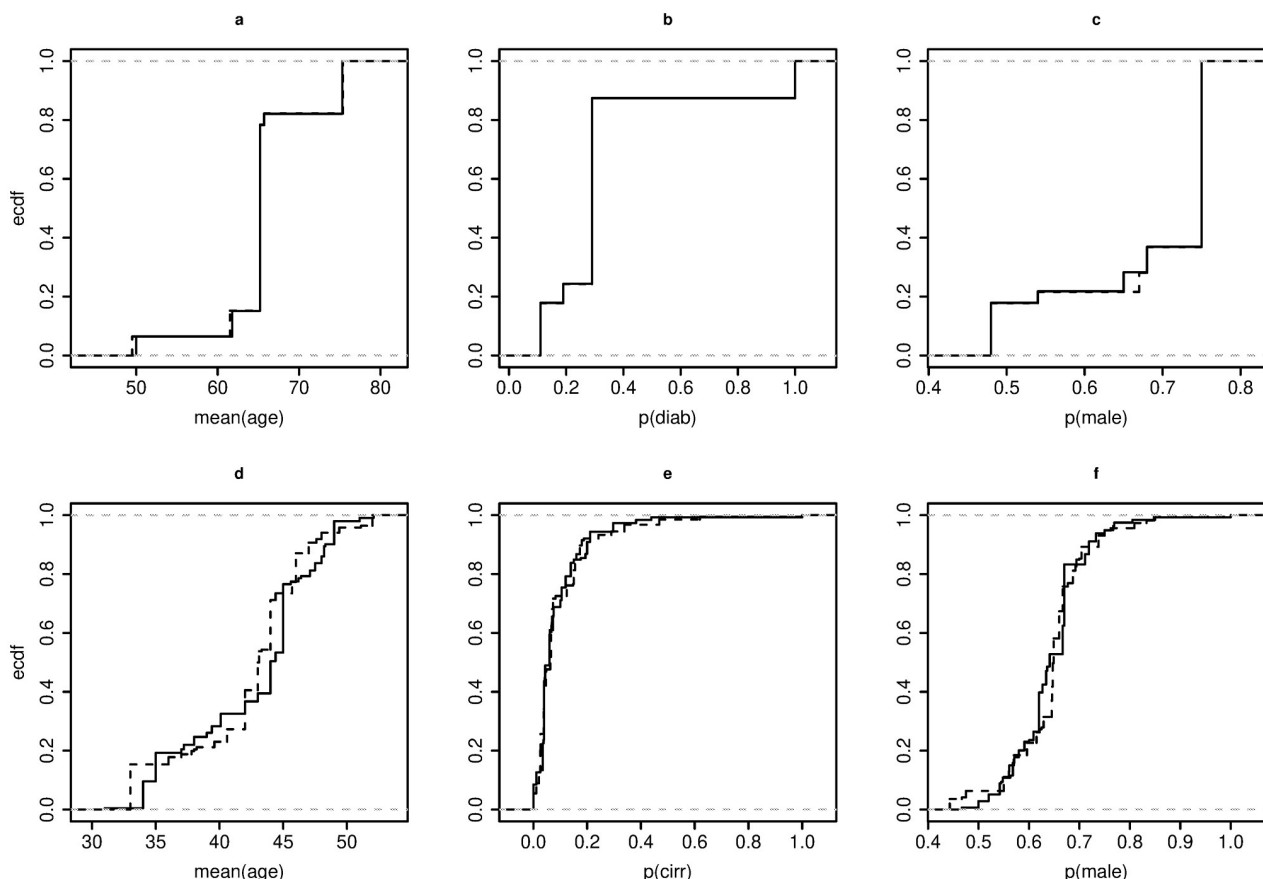

**Fig 5. ECDFs of the *exp* (——) and *ctrl* (——) meta-arms in the European studies, for the PLVs.** BAL1 (a, b, c) of the *Chol* dataset and BAL2 (d, e, f) of the *Hep* dataset.

as follows:

$$arm_k = \begin{cases} k = 0 \text{ if } ctrl \\ k = 1 \text{ if } exp \end{cases}$$

$$imb_m = \begin{cases} m = 0 \begin{cases} \text{if } \mathbf{BAL1} \in Chol \\ \text{if } \mathbf{BAL2} \in Hep \end{cases} \\ m = 1 \begin{cases} \text{if } \mathbf{UNB1} \in Chol \\ \text{if } \mathbf{UNB2} \in Hep \end{cases} \end{cases}$$

The model is defined as $logit(p_{km}) = \beta_0 + \beta_1 arm_k + \beta_2 imb_m$, where $k = 0, 1$ and $m = 0, 1$. The parameter estimates and their standard errors are reported in Table 4.

It is evident that *imb* has a significant effect on the outcomes, while the interactions are not significant. This suggests that the presence of unbalanced trials should always be investigated, to avoid biased estimates of treatment effects. It is important to emphasize that logistic regression with the inclusion of a dummy variable indicating the presence of unbalanced studies does not resolve the problem when the sample size of balanced studies obtained through the procedure is limited. Instead, it serves as a warning about the effect size of both balanced and unbalanced studies on the outcome.

## Conclusions

As highlighted in the introduction, there is a lack of statistical methods for assessing systematic differences in patients' characteristics in meta-analysis studies, even though numerous methods and procedures exist for correcting covariate imbalances in individual RCTs [11, 19]. It is important to note that incorporating unbalanced trials can have a significant effect on the assessment of the response [90]. In this context, we conducted a meta-regression aiming at illustrating the effect that including balanced trials would have had on the outcome. In fact, the meta-regression equation tells us just that the presence of unbalanced trials may change (if the parameter is significant) the effect size.

In clinical practice, researchers always encounter trials with multiple covariates, and it becomes essential to evaluate whether a trial can be considered balanced as a whole, regardless of the balance of individual covariates. In this regard, to address this issue, we presented a method for removing trials that simultaneously considers three covariates, building a prior study [22] that tackled the issue in the presence of a single covariate. The method involves constructing meta-arms, which are collections of similar randomized experimental or control arms. These meta-arms are then compared through their ECDFs to determine whether the randomization concerning a set of risk factors holds. If randomization is not upheld, the trials responsible for the imbalance are identified iteratively using a statistical test based on the

**Table 4. Estimated parameters of the meta-regression logistic models.** S1 (*Chol* dataset) and S2 (*Hep* dataset).

| Coefficients | S1 | | | S2 | | |
|---|---|---|---|---|---|---|
| | Estimate | Std.Err. | *p* | Estimate | Std.Err. | *p* |
| intercept | -1.502 | 0.018 | < 0.001 | -1.942 | 0.059 | < 0.001 |
| *arm* | -0.301 | 0.023 | < 0.001 | 1.344 | 0.068 | < 0.001 |
| *imb* | -0.427 | 0.023 | < 0.001 | -0.139 | 0.069 | < 0.05 |

distance between the ECDFs. We have also conducted a simulation study with various scenarios to strengthen to the method's validity.

One limitation of this method is that it may lead to a reduction of the number of trials involved in the meta-analysis. Therefore, investigators must decide whether the meta-analysis is still meaningful after the removal of many unbalanced trials.

Finally, our work proposes a method of backward elimination of studies. Nowadays, meta-analyses have the potential to include many studies, so the removal of trials should not compromise the conduct of the meta-analysis itself. However, there are alternative statistical methods, such as propensity score methods, which address imbalance through re-weighting procedures and could offer a solution. Nonetheless, these methods are generally more computationally intensive, and employ a distinct approach.

## Supporting information

**S1 Appendix. Simulations.**
(PDF)

**S2 Appendix. Meta-arms and ECDF.**
(PDF)

**S1 File. PRISMA checklist.**
(PDF)

**S2 File. The studies' selection procedure for the *Hep* dataset.**
(PDF)

**S3 File. *Chol* dataset.** File of the *Chol* dataset.
(TXT)

**S4 File. *Hep* dataset.** File of the *Hep* dataset.
(TXT)

**S1 Table. Ribavirin plus interferon versus interferon for chronic hepatitis C's 40 studies (*Hep* dataset) selected SLV and PLVs.**
(XLSX)

**S2 Table. European and Non-European studies in *Chol* and *Hep* datasets.**
(XLSX)

**S3 Table. 1st iteration: Anderson-Darling test statistics $T_{1(-i)}$, $T_{2(-i)}$, $T_{3(-i)}$, and $T_{c(-i)}$.** *Chol* dataset.
(XLSX)

**S4 Table. 2nd iteration: Anderson-Darling test statistics $T_{1(-i)}$, $T_{2(-i)}$, $T_{3(-i)}$, and $T_{c(-i)}$.** *Chol* dataset.
(XLSX)

**S5 Table. 6th iteration: Anderson-Darling test statistics $T_{1(-i)}$, $T_{2(-i)}$, $T_{3(-i)}$, and $T_{c(-i)}$.** *Chol* dataset.
(XLSX)

## Acknowledgments

We would like to thank Vincenzo Giuseppe Genova, Vito Michele Rosario Muggeo, and Michele Tumminello for their useful suggestions.

## Author Contributions

**Conceptualization:** Massimo Attanasio.

**Data curation:** Fabio Aiello, Fabio Tinè.

**Formal analysis:** Massimo Attanasio, Fabio Aiello.

**Funding acquisition:** Massimo Attanasio.

**Methodology:** Massimo Attanasio, Fabio Aiello.

**Software:** Fabio Aiello.

**Supervision:** Massimo Attanasio, Fabio Aiello, Fabio Tinè.

**Writing – original draft:** Massimo Attanasio, Fabio Aiello, Fabio Tinè.

**Writing – review & editing:** Massimo Attanasio, Fabio Aiello.

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
