## [Decision Letter · Decision Letter 0]

15 Aug 2023

PONE-D-23-10166A statistical method to assess and to adjust for covariate imbalance in meta-analysis.PLOS ONE

Dear Dr. Aiello,

Thank you for submitting your manuscript to PLOS ONE. After careful consideration, we feel that it has merit but does not fully meet PLOS ONE’s publication criteria as it currently stands. Therefore, we invite you to submit a revised version of the manuscript that addresses the points raised during the review process.

Important comments have been made and even strong doubts have been expressed by the Reviewers. Please consider all these issues carefully.

Specifically, the crucial question has been raised whether there is enough new material in the present paper to satisfy the PLOS ONE publication criteria. Please refer to the following webpage:

https://journals.plos.org/plosone/s/criteria-for-publication

In particular, consider the paragraph about Replication Studies: "If a submitted study replicates or is very similar to previous work, authors must provide a sound scientific rationale for the submitted work and clearly reference and discuss the existing literature. Submissions that replicate or are derivative of existing work will likely be rejected if authors do not provide adequate justification."

We look forward to receiving your revised manuscript.

Kind regards,

Harald Heinzl

Academic Editor

PLOS ONE

Journal Requirements:

Reviewers' comments:

Reviewer's Responses to Questions

**Comments to the Author**

1. Is the manuscript technically sound, and do the data support the conclusions?

Reviewer #1: Yes

Reviewer #2: Partly

Reviewer #3: Yes

2. Has the statistical analysis been performed appropriately and rigorously? 

Reviewer #1: Yes

Reviewer #2: Yes

Reviewer #3: Yes

3. Have the authors made all data underlying the findings in their manuscript fully available?

Reviewer #1: Yes

Reviewer #2: Yes

Reviewer #3: No

4. Is the manuscript presented in an intelligible fashion and written in standard English?

Reviewer #1: Yes

Reviewer #2: Yes

Reviewer #3: Yes

5. Review Comments to the Author

Reviewer #1: This paper presents a novel method for identifying unbalanced trials in meta-analysis, with the aim of achieving combinability. This concept holds significant importance and has been frequently discussed in meta-analysis literature. The proposed approach relies on a backward reduction procedure utilizing the combined Anderson-Darling test, enabling efficient detection and elimination of unbalanced trials. Moreover, the method can be seen as an extension of the one proposed by Aiello, Attanasio, and Tinè (2011). I have some comments on this paper, please see them in the attachment.

Reviewer #2: The paper touches on the important issue of combinability in meta-analysis, in particular related to covariate (im)balance within trials. A method for identifying unbalanced trials in a meta-analysis is proposed. This paper seems to expand on previous work (Statistics in Medicine 2011), with a new automated procedure for identifying a subset of unbalanced (up to 3 covariates) trials within a meta-analysis. It was interesting to read. My questions and comments can be found below.

Regarding the novelty of the work

- The method described in this paper was introduced in the previous paper "Assessing covariate imbalance in meta-analysis studies", Stat. Med. 2011. Some of the contents are similar, as well as the example datasets. I think that the proposed algorithm for detecting trials that contribute to imbalance is a welcome expansion of the previously described methodology. I am not yet fully convinced that the proposed meta-regression provides a way to adjust for detected imbalances. Please could the editor advise as to whether there is enough new material in the present paper to satisfy the PLOS ONE publication criteria?

- Quite a few references are not very recent and similar to the references of the previous paper. Perhaps some more recent references would help to provide context?

Regarding technical aspects of the work

- Imbalance is assessed for summary statistics of a covariate within study arms, f.i., the mean. Does it matter that other aspects of the distribution are not taken into account in the imbalance assessment?

- The distribution of summary statistics over all control arms is compared to the distribution over all experimental arms. The link between two arms of the same trial is broken in this way, while respecting within-trial comparisons is usually viewed as important in the meta-analysis of the outcome. Please clarify whether this has any effect on the interpretation of the results.

- Please could you add some information about the number of studies needed in a meta analysis to reliably estimate the ECDFs of interest/have enough power for the nonparametric comparison?

- The meta-regression is introduced as a way to adjust for baseline imbalance. However, in the section itself, the goal of the meta-regression is formulated as: "to evaluate whether the treatment's effect (i.e., the arm type) on the outcome varies when controlling for these imbalances." So this is more of a detection/evaluation of imbalances than an adjustment. Could you explain which adjustment for baseline imbalances you had in mind based on the meta-regression?

- The goal as stated is "to evaluate whether the treatment’s effect (i.e., the arm type) on the outcome varies when controlling for these imbalances." I am not sure that the proposed regression equation satisfies this goal. To me, a significant coefficient of 'imb' in this equation would mean that the overall outcome is different in one group of studies vs the rest of the studies in the meta-analysis; I do not immediately see how it says anything about variation in treatment effect. Could you please explain how this regression model detects an effect of within-study imbalances on a treatment effect?

- The conclusion of the adjustment section is unclear to me: "It is noticeable that dummy imb yields a significative effect on the outcomes"--yes. "This means that the presence of imbalance between the meta-arms should be always investigated and eventually included, to avoid biased estimates of the treatments' effects."--How would this presence be included? And how does this conclusion follow from the results in this section?

- A large part of the conclusion section repeats the study motivation from the introduction. On the other hand, I was missing some reflections on/implications of the results. In my view, this section could be improved by shortening the first 3 paragraphs and expanding the reflections on the results, the limitations of the study and the possible implications of this work.

- "Adjust for covariate imbalance" is part of the title. In the paper, I have only found methods to evaluate covariate imbalance in meta-analysis. Please indicate where an actual adjustment is described, or adapt the wording of the title.

Regarding the use of English

- The article is written in intelligible English, however there are some minor errors throughout. For example:"responsible of" instead of "responsible for", "denature the meta-analysis", "from which we excluded 72 of them" instead of "of which we excluded 72", "significative" instead of "significant". I would recommend having the paper reviewed by a native English speaker to make sure everything is correct.

Reviewer #3: The paper describes a new approach for assessing the study combinability in a meta-analysis. This is an important topic but several clarifications are needed:

1. The paper aims to establish combinability from the angle of covariate imbalance. However, combinability, as the author wrote, focuses on “the extent to which separate studies measure the same thing” whereas covariate imbalance between arms or meta-arms is interested in the similarity between arms rather than studies. It would be better to have more explanation in the introduction for why the similarity between studies (combinability) is violated if there is dissimilarity between arms (covariate imbalance).

2. Does the proposed method only apply to meta-analysis with individual participant data (IPD)? It seems that one would need to know the patient-level variables in the IPD meta-analysis to use the proposed method.

3. How the Anderson-Darling criterion is related to comparison between studies in a meta-analysis need be clarified. Did the authors pool all samples from arm k across all studies in calculating the k-sample Anderson-Darling criterion?

4. In a real met-analysis, some studies may not include certain covariates that are present in other studies. How the proposed method can handle missing PLVs or SLVs in specific studies is not discussed.

5. Is type I error controlled when assessing the basic combinability and identifying the unbalanced trials?

6. In the procedure of identifying the unbalanced trials, the minimum of test statistic no longer has the same distribution as the test statistic according to the extreme value theory.

7. When adjusting for the effect of imbalance in the meta-regression, what’s the reason that there is not an interaction between indicator for imbalance and treatment arms?

Minor

1. The mean of A_{hk}^2 is said to be k-1. Do authors have any reference or derivation for this?

2. Line 89, Page 4: “RCTs” rather than “RTCs”.

3. Line 111, Page 5: The authors indicated that this paper started from a previous work in reference 19, but line 84 in page 4 suggested that reference 17 has done similar work too. The authors may need to add reference 19 in the introduction to suggest the difference between reference 17 and 19.

6. PLOS authors have the option to publish the peer review history of their article (what does this mean?). If published, this will include your full peer review and any attached files.

Reviewer #1: **Yes: **Ming Zhang

Reviewer #2: No

Reviewer #3: No

---

## [Author Response · Author response to Decision Letter 0]

15 Oct 2023

PLOS ONE Decision: Revision required [PONE-D-23-10166]

Reviewers' comments:

Reviewer's Responses to Questions

Reviewers' comments:

Reviewer's Responses to Questions

Comments to the Author

1. Is the manuscript technically sound, and do the data support the conclusions?

Reviewer #1: Yes

Reviewer #2: Partly

Reviewer #3: Yes

ANSWER: We now conducted a simulation (see Appendix B) to give more strength to our proposal. 

2. Has the statistical analysis been performed appropriately and rigorously?

Reviewer #1: Yes

Reviewer #2: Yes

Reviewer #3: Yes

3. Have the authors made all data underlying the findings in their manuscript fully available?

The PLOS Data policy requires authors to make all data underlying the findings described in their manuscript fully available without restriction, with rare exception (please refer to the Data Availability Statement in the manuscript PDF file). The data should be provided as part of the manuscript or its supporting information or deposited to a public repository. For example, in addition to summary statistics, the data points behind means, medians and variance measures should be available. If there are restrictions on publicly sharing data—e.g., participant privacy or use of data from a third party—those must be specified.

Reviewer #1: Yes

Reviewer #2: Yes

Reviewer #3: No

ANSWER: We uploaded the csv files of the Chol and Hep datasets in the Supporting Information.

4. Is the manuscript presented in an intelligible fashion and written in standard English? 

Reviewer #1: Yes

Reviewer #2: Yes

Reviewer #3: Yes

Review Comments to the Author

Please use the space provided to explain your answers to the questions above. You may also include additional comments for the author, including concerns about dual publication, research ethics, or publication ethics. (Please upload your review as an attachment if it exceeds 20,000 characters).

Reviewer #1: This paper presents a novel method for identifying unbalanced trials in meta-analysis, with the aim of achieving combinability. This concept holds significant importance and has been frequently discussed in meta-analysis literature. The proposed approach relies on a backward reduction procedure utilizing the combined Anderson-Darling test, enabling efficient detection and elimination of unbalanced trials. Moreover, the method can be seen as an extension of the one proposed by Aiello, Attanasio, and Tinè (2011). I have some comments on this paper, please see them in the attachment.

Comments:

This paper presents a novel method for identifying unbalanced trials in meta-analysis, with the aim of achieving combinability. This concept holds significant importance and has been frequently discussed in meta-analysis literature. The proposed approach relies on a backward reduction procedure utilizing the combined Anderson-Darling test, enabling efficient detection and elimination of unbalanced trials. Moreover, the method can be seen as an extension of the one proposed by Aiello, Attanasio, and Tinè (2011).

ANSWER 1: 

The procedure proposed in this work is a “necessary” extension of the previous work (Aiello, Attanasio, Tiné, 2011) for several reasons. In fact, the main differences compared to the previous work are:

1. the extension of the combinability procedure in the presence of three covariates, considering that clinical studies often involve more than one covariate. This also led to a generalization of the test statistics used (for a better understanding of this aspect, an extra sentence has been inserted in the introduction).

2. the inclusion of a new section (“Adjusting for the effect of the imbalance on the outcome”), a kind of ex-post verification, dedicated to estimating the effect size with unbalanced and balanced trials.

3. the inclusion of a simulation in the appendix (as suggested by reviewer 1), which adds more strength to the procedure.

These 3 specifications are added in the Section “The proposed statistical method”.

Major Comments:

QUESTION 2:

• The original paper and the previous work by Aiello et al. used the notation of AkN. It will be better if you either use this fashion to write your formula here (e.g, adding one more subscript here to represent hth PLV) or write your A2 explicitly if you mean the different thing.

ANSWER 2: 

Thank you. For the sake of clarity, we added the extra subscript N to the statistics (from row 214 to row 232)

QUESTION 3:

• The k-sample Anderson-Darling criterion assumes independent samples. However, it is possible that we can encounter dependent studies (See Lin and Sullivan (2009) and Han et al. (2016)). Can your method address this scenario? Any comments or discussions will be appreciated.

ANSWER 3: 

Our method does not address the scenario of dependent samples. We just added an extra sentence at the end of section “Notation”: “In the case of dependent samples, one can refer to the suggestions made by Lin and Sullivan and Han et al. (2009)”. 

QUESTION 4:

Any statistical properties about A2c or Tc? For example, what kind of distributions will they follow?

ANSWER 4: 

Thank you for this suggestion. In section “NOTATION”, we added the sentence “Details on the statistical distributions are in [43]”. 

QUESTION 5:

• On page 16, it is evident that the results obtained are significantly different. However, since the true effect sizes of the datasets are unknown, it is challenging to draw definitive conclusions. To establish the efficacy of the proposed process and demonstrate its potential in yielding more reliable results, I suggest conducting several small simulation studies. These simulations can assess various aspects, such as the accuracy of effect size estimation, reduction in variance, and other relevant metrics.

ANSWER 5:

Thank you. We conducted the simulation study with 9 different scenarios with just one covariate (see Appendix B)

QUESTION 6:

The backward reduction procedure is a very interesting method for detecting biased studies within a meta-analysis, where Tc statistics will be used as a criterion. However, when there is a limited number of studies available, the reliability of Tc statistics may decrease due to reduced statistical power. Do you have any comments or discussions about this point?

ANSWER 6.1:

Thank you for this issue. In the conclusions, we already wrote that “A limitation of this method is the eventual reduction of the trials involved in the meta-analysis”, but we added the following sentence at the very end: 

“Finally, our work proposes a method of backward elimination of studies. Nowadays, meta-analyses have the potential to include many studies, and the proposed method should be capable of ensuring a "sufficient" number of studies. However, there are also other statistical methods, such as propensity score, which address imbalance through re-weighting procedures that could provide a solution, that is certainly more computationally intensive with a completely different approach”.

ANSWER 6.2:

Moreover, we also added at the end of the section “Adjusting for the effect of the Imbalance in the outcome” the sentences: 

“It's important to emphasize that logistic regression with the inclusion of a dummy variable indicating the presence of unbalanced studies does not resolve the problem when the sample size of balanced studies obtained through the procedure is limited. It serves as a warning about the impact of both balanced and unbalanced studies on the outcome.”

QUESTION 7:

• The primary focus of this paper lies in studies with higher incidence rates, which typically involve more common events. However, it is essential to consider the potential impact of the proposed method on rare events as well, where background incidence rates can be lower, even below 0.5%. See Bhaumik et al. (2012) and Zhang et al. (2023).

ANSWER 7:

Our paper does not address meta-analysis for rare events. At the beginning of the section “Two illustrative examples”, we added the sentence “The examples used in this work refer to meta-analyses with higher incidence rates”

Minor Comments:

QUESTION 8:

• I can’t find a definition of A2h.

ANSWER 8:

Thanks. We modified it (see ANSWER 2)

QUESTION 9:

• I have trouble finding Table 2-4, please check the label indexes.

ANSWER 9:

Thank you, it was not clear. In Table 2, we changed the caption and some labels. In Table 3, we changed the captions and some labels. In Table 3, we changed the captions and some labels. 

In Table 4, we changed the captions and some labels. We changed the captions of the figures too. In the final manuscript the changes are marked.

QUESTION 10:

• Any comments on Fig 4?

ANSWER 10:

Yes, we forgot to write the comment. We added (page 15) now “In fig.4, the ECDFs for the first dataset (a, b, and c) show closeness, while in the second dataset (d, e, and f), the ECDFs are less close. This is probably due the different distribution among the studies in the datasets”

QUESTION 11:

• I like the tables (Table I & II) in Aiello, Attanasio, and Tinè (2011) to explain how ECDF was calculated. Can you include some similar tables here? You may include them in the Appendix.

ANSWER 11:

Yes, we did. See Appendix A.

References

- Aiello, Fabio, Massimo Attanasio, and Fabio Tinè (Sept. 2011). “Assessing covariate imbalance in meta-analysis studies”. en. In: Statistics in Medicine 30.22, pp. 2671–2682.

- Bhaumik, Dulal K. et al. (June 2012). “Meta-Analysis of Rare Binary Adverse Event Data”. en. In: Journal of the American Statistical Association 107.498, pp. 555–567.

- Han, Buhm et al. (May 2016). “A general framework for meta-analyzing dependent 

studies with overlapping subjects in association mapping”. en. In: Human Molecular Genetics 25.9, pp. 1857–1866.

- Lin, Dan-Yu and Patrick F. Sullivan (Dec. 2009). “Meta-Analysis of Genomewide Association Studies with Overlapping Subjects”. In: The American Journal of Human Genetics 85.6, pp. 862–872.

- Zhang, Ming et al. (May 2023). “Bayesian estimation and testing in randomeffects meta-analysis of rare binary events allowing for flexible group variability”. In: Statistics in Medicine 42.11, pp. 1699–1721.

Reviewer #2

The paper touches on the important issue of combinability in meta-analysis, in particular related to covariate (im)balance within trials. A method for identifying unbalanced trials in a meta-analysis is proposed. This paper seems to expand on previous work (Statistics in Medicine 2011), with a new automated procedure for identifying a subset of unbalanced (up to 3 covariates) trials within a meta-analysis. It was interesting to read. My questions and comments can be found below.

Regarding the novelty of the work

We split the questions into small questions 

QUESTION 1:

- The method described in this paper was introduced in the previous paper "Assessing covariate imbalance in meta-analysis studies", Stat. Med. 2011. Some of the contents are similar, as well as the example datasets. I think that the proposed algorithm for detecting trials that contribute to imbalance is a welcome expansion of the previously described methodology. I am not yet fully convinced that the proposed meta-regression provides a way to adjust for detected imbalances. 

ANSWER 1:

Thank you, we really appreciate your comment. We wrote a misleading title for the section "adjusting for the effect of the imbalance on the outcome variable." This section is somewhat of a digression from the method proposed to identify unbalanced trials. The proposed solution is simply to eliminate the unbalanced trials. Therefore, we have changed the section title to "the effect of the imbalance on the outcome variable." The application of meta-regression only serves to illustrate the effect that including balanced trials would have had on the outcome. We certainly were not very clear, so we have completely changed this section (see the new version).

QUESTION 2:

Please could the editor advise as to whether there is enough new material in the present paper to satisfy the PLOS ONE publication criteria?

ANSWER 2:

Editor will answer

QUESTION 3:

- Quite a few references are not very recent and similar to the references of the previous paper. Perhaps some more recent references would help to provide context?

ANSWER 3:

Thank you for the suggestion. We have reviewed the references and made the following updates:

- We have included two additional papers in the reference list: Wewege et al. (2022) in Hypertension Research (45: 1643-1652) and Hicks et al. in the Journal of Clinical Epidemiology (2016). Both of these papers address baseline imbalance and utilize statistical methods to assess covariate differences, removing studies where the differences are unacceptable. These references have been included into the paper.

- We have also added another reference by Clark et al. (2014 and 2015 J Clinl Epidemiology, and 2016 BMI). These papers discuss baseline heterogeneity resulting from incorrect allocation concealment. The 2016 reference has been included in the manuscript, highlighting the connection between allocation concealment and its potential impact on selection bias in patient assignment to intervention groups.

These modifications have been made in the Introduction, lines 80-90.

Regarding technical aspects of the work

QUESTION 4:

- Imbalance is assessed for summary statistics of a covariate within study arms, f.i., the mean. Does it matter that other aspects of the distribution are not taken into account in the imbalance assessment?

ANSWER 4:

Very few meta-analyses report other statistics as standard deviations or other statistics. The sample size is usually important.

QUESTION 5:

- The distribution of summary statistics over all control arms is compared to the distribution over all experimental arms. The link between two arms of the same trial is broken in this way, while respecting within-trial comparisons is usually viewed as important in the meta-analysis of the outcome. Please clarify whether this has any effect on the interpretation of the results.

ANSWER 5:

Yes, you are right, the link between two arms of the same trial is broken in this way, in fact we construct two meta-arms (an exp meta-arm and a ctrl meta-arm). Indeed, we are not interested to the outcome (see ANSWER 1)

QUESTION 6:

- Please could you add some information about the number of studies needed in a metanalysis to reliably estimate the ECDFs of interest/have enough power for the nonparametric comparison?

ANSWER 6:

This is a good question that arises frequently. ECDFs step functions. Typically, each step corresponds to one study, so having fewer than 10 studies is not advisable. 

QUESTION 7:

- The meta-regression is introduced as a way to adjust for baseline imbalance. However, in the section itself, the goal of the meta-regression is formulated as: "to evaluate whether the treatment's effect (i.e., the arm type) on the outcome varies when controlling for these imbalances." So, this is more of a detection/evaluation of imbalances than an adjustment. Could you explain which adjustment for baseline imbalances you had in mind based on the meta-regression?

ANSWER 7:

Yes, you are completely right, it is only a detection/evaluation. We changed the title of the section and all the Section. We think that we already answered to this question at #1 

QUESTION 8:

- The goal as stated is "to evaluate whether the treatment’s effect (i.e., the arm type) on the outcome varies when controlling for these imbalances." I am not sure that the proposed regression equation satisfies this goal. To me, a significant coefficient of 'imb' in this equation would mean that the overall outcome is different in one group of studies vs the rest of the studies in the meta-analysis; I do not immediately see how it says anything about variation in treatment effect. Could you please explain how this regression model detects an effect of within-study imbalances on a treatment effect?

ANSWER 8:

Yes, you are correct. We have realized that there was a misunderstanding. Your observation that "I do not immediately see how it says anything about variation in treatment effect" is correct. The meta-regression equation reveals only that the presence of unbalanced trials might alter the effect size (if the parameter is significant). We hope that these explanations are satisfactory, and we trust that the new section is as well.

QUESTION 9:

- The conclusion of the adjustment section is unclear to me: "It is noticeable that dummy imb yields a significative effect on the outcomes" -- yes. "This means that the presence of imbalance between the meta-arms should be always investigated and eventually included, to avoid biased estimates of the treatments' effects." -- How would this presence be included? And how does this conclusion follow from the results in this section?

ANSWER 9:

This sentence is just a consequence of the previous mistake (see Answers 1,7, and 8)

QUESTION 10:

- A large part of the conclusion section repeats the study motivation from the introduction. On the other hand, I was missing some reflections on/implications of the results. In my view, this section could be improved by shortening the first 3 paragraphs and expanding the reflections on the results, the limitations of the study and the possible implications of this work.

ANSWER 10:

We reduced the first three paragraphs of the conclusions, and we expanded the reflections of the results. See the new manuscript

 QUESTION 11:

- "Adjust for covariate imbalance" is part of the title. In the paper, I have only found methods to evaluate covariate imbalance in meta-analysis. Please indicate where an actual adjustment is described or adapt the wording of the title.

ANSWER 11:

We changed the wording of the title. The new one is: A Statistical Method for Removing Unbalanced Trials with Multiple Covariates in Meta-Analysis.

QUESTION 12:

Regarding the use of English

- The article is written in intelligible English, however there are some minor errors throughout. For example: "responsible of" instead of "responsible for", "denature the meta-analysis", "from which we excluded 72 of them" instead of "of which we excluded 72", "significative" instead of "significant". I would recommend having the paper reviewed by a native English speaker to make sure everything is correct.

ANSWER 12:

Thank you for your suggestions. We modified accordingly. A native English speaker read it.

Reviewer #3

QUESTION 1

The paper describes a new approach for assessing the study combinability in a meta-analysis. This is an important topic, but several clarifications are needed:

The paper aims to establish combinability from the angle of covariate imbalance. However, combinability, as the author wrote, focuses on “the extent to which separate studies measure the same thing” whereas covariate imbalance between arms or meta-arms is interested in the similarity between arms rather than studies. It would be better to have more explanation in the introduction for why the similarity between studies (combinability) is violated if there is dissimilarity between arms (covariate imbalance).

ANSWER 1. Yes, we tried to explain better 

Previous version. Row 77

The underlying assumption in meta-analysis is that, under the process of random subject allocation to the experimental (exp) and control (ctrl) arms, the expected level of imbalance in covariate distribution is zero [14]. But, summing up individual RCTs with insignificant level of imbalance may cause a significant covariate imbalance in meta-trials.

Only a few papers have investigated the covariate imbalance that occurs in meta-analysis studies. Trowman et al. [15] state that a meta-analysis imbalance may not result just from a baseline imbalance of one particular trial, but rather from a cumulative effect of smaller imbalances.

New version: row 77

In essence, the issue is that the similarity among studies, known as "combinability", is violated when dissimilarity arises between the experimental (exp) and control (ctrl) arms due to covariate imbalances. Fundamentally, meta-analysis operates on the premise that, during the random allocation process to the exp and control (ctrl) arms, the expected level of covariate distribution imbalance should ideally be zero [14]. The covariate balance is not always checked before conducting a meta-analysis, automatically assuming that individual studies are all balanced. However, it can happen that some studies do not exhibit covariate imbalance for some or all covariates, or, as Trowman et al, [15] state that a meta-analysis imbalance may not result just from a baseline imbalance of one trial, but rather from a cumulative effect of smaller imbalances. In both cases, the meta-analysis present covariate imbalance"

QUESTION 2:

2. Does the proposed method only apply to meta-analysis with individual participant data (IPD)? It seems that one would need to know the patient-level variables in the IPD meta-analysis to use the proposed method.

ANSWER 2:

The terminology used for patient-level variables was employed solely to distinguish them from study-level variables. The proposed method does not apply to Individual Patient Data (IPD) meta-analysis.

QUESTION 3:

3. How the Anderson-Darling criterion is related to comparison between studies in a meta-analysis need be clarified. Did the authors pool all samples from arm k across all studies in calculating the k-sample Anderson-Darling criterion?

ANSWER 3:

Yes, we pooled the meta-arms over the 3 covariates. We modified the subscripts of the A-D statistics for k samples in the section “Notation”. In section “NOTATION”, we added the sentence “Details on the statistical distributions are in [43]”. 

QUESTION 4:

4. In a real metanalysis, some studies may not include certain covariates that are present in other studies. How the proposed method can handle missing PLVs or SLVs in specific studies is not discussed.

ANSWER 4:

We are not concerned with this issue in this paper. We may suggest usual methods of missing values imputation based on other covariates. 

QUESTION 5:

5. Is type I error controlled when assessing the basic combinability and identifying the unbalanced trials?

ANSWER 5:

Thank you , we forgot to specify in the iteration procedure that alfa=0.05. We modified it.

QUESTION 6:

6. In the procedure of identifying the unbalanced trials, the minimum of test statistic no longer has the same distribution as the test statistic according to the extreme value theory.

ANSWER 6:

Schotz and Stephens (1997) reports for the A-D test that: “It appears that the proposed tests maintain their levels quite well even for samples as small as ni = 5. For small sample sizes the observed levels tend to be slightly conservative, that is, smaller than nominal, for extreme tail probabilities”

QUESTION 7:

7. When adjusting for the effect of imbalance in the meta-regression, what’s the reason that there is not an interaction between indicator for imbalance and treatment arms?

ANSWER 7:

The proposed method aims to exclude studies that exhibit covariate imbalance. It is plausible that the covariates used may also be significant risk factors for the outcome, but the interaction between the arm and the imbalance indicator may or may not be present. We had already investigated the logistic models with interactions. Here the results (gr=treatment; imb= inbalance; hep= dataset Hep; Chol= dataset Chol):

For the sake of brevity, we did not include these results.

Minor

QUESTION 8:

1.The mean of A_{hk}^2 is said to be k-1. Do authors have any reference or derivation for this?

ANSWER 8:

Schotz and Stephens (1997) reports the mean of A_{hk}^2 at page 919

QUESTION 9:

2. Line 89, Page 4: “RCTs” rather than “RTCs”.

ANSWER 9:

Yes, thank you

QUESTION 10:

3. Line 111, Page 5: The authors indicated that this paper started from a previous work in reference 19, but line 84 in page 4 suggested that reference 17 has done similar work too. The authors may need to add reference 19 in the introduction to suggest the difference between reference 17 and 19.

ANSWER 9:

Another referee asked us to update the references of the last years. So, we added some more recent papers. However, reference [19] Aiello, Attanasio, and Tiné represents our previous work.

---

## [Decision Letter · Decision Letter 1]

21 Nov 2023

A statistical method for removing unbalanced trials with multiple covariates in meta-analysis

PONE-D-23-10166R1

Dear Dr. Aiello,

We’re pleased to inform you that your manuscript has been judged scientifically suitable for publication and will be formally accepted for publication once it meets all outstanding technical requirements.

Kind regards,

Harald Heinzl

Academic Editor

PLOS ONE

Additional Editor Comments:

Two remarks:

line 85: "with" instead of "wth"

lines 117-118: The sentence "for a better understanding of this aspect, changes have been made in the introduction" sounds like an answer to a reviewer which, of course, should not appear in the paper. I suggest that you reformulate this sentence.

Reviewers' comments:

Reviewer's Responses to Questions

**Comments to the Author**

1. If the authors have adequately addressed your comments raised in a previous round of review and you feel that this manuscript is now acceptable for publication, you may indicate that here to bypass the “Comments to the Author” section, enter your conflict of interest statement in the “Confidential to Editor” section, and submit your "Accept" recommendation.

Reviewer #1: All comments have been addressed

Reviewer #2: All comments have been addressed

2. Is the manuscript technically sound, and do the data support the conclusions?

Reviewer #1: Yes

Reviewer #2: (No Response)

3. Has the statistical analysis been performed appropriately and rigorously? 

Reviewer #1: Yes

Reviewer #2: (No Response)

4. Have the authors made all data underlying the findings in their manuscript fully available?

Reviewer #1: Yes

Reviewer #2: (No Response)

5. Is the manuscript presented in an intelligible fashion and written in standard English?

Reviewer #1: Yes

Reviewer #2: (No Response)

6. Review Comments to the Author

Reviewer #1: (No Response)

Reviewer #2: (No Response)

7. PLOS authors have the option to publish the peer review history of their article (what does this mean?). If published, this will include your full peer review and any attached files.

Reviewer #1: No

Reviewer #2: No

---

## [Editor Report · Acceptance letter]

6 Dec 2023

PONE-D-23-10166R1 

A statistical method for removing unbalanced trials with multiple covariates in meta-analysis 

Dear Dr. Aiello:

I'm pleased to inform you that your manuscript has been deemed suitable for publication in PLOS ONE. Congratulations! Your manuscript is now with our production department. 

Kind regards, 

on behalf of

Dr. Harald Heinzl 

Academic Editor

PLOS ONE